# Optic Nerve Ultrasound Evaluation in Children: A Review

**DOI:** 10.3390/diagnostics13030535

**Published:** 2023-02-01

**Authors:** Giulia Abbinante, Livio Vitiello, Alessia Coppola, Giulio Salerno, Vincenzo Gagliardi, Alfonso Pellegrino

**Affiliations:** Eye Unit, “Luigi Curto” Hospital, Azienda Sanitaria Locale Salerno, Polla, 84035 Salerno, Italy

**Keywords:** children, optic nerve sheath diameter, ONSD, POCUS, ultrasound

## Abstract

Managing patients with neurocritical illness requires monitoring and treating elevated intracranial pressure (ICP), especially in cases in children. In terms of precise and real-time measurements, invasive ICP measurements are presently the gold standard for the initial diagnosis and follow-up ICP assessments. As a rapid and non-invasive way to detect elevated ICP, point-of-care ultrasonography (POCUS) of optic nerve sheath diameter (ONSD) has been proposed. The utility of bedside POCUS of ONSD to detect elevated ICP with excellent diagnostic test accuracy in adults has already been demonstrated. Nonetheless, data on the relationship between POCUS of ONSD and ICP in children are scarce. Therefore, the purpose of this review is to point out the most recent findings from the pediatric published literature and briefly discuss what was assessed with ONSD ultrasound examination, and also to describe and discuss the diagnostic procedures available for optic nerve ultrasound appraisal. A search of the medical databases PubMed and Scopus was carried out. The terms such as “ocular ultrasonography”, “ICP assessment”, “children”, “point-of-care ultrasound”, and “POCUS” were searched. In conclusion, the use of the standardized A-scan technique coupled with the B-scan technique should be suggested to provide data that are as accurate, precise, repeatable, and objective as possible.

## 1. Introduction

Poor neurological outcomes following a traumatic brain injury have been linked to increased intracranial pressure (ICP), which is a significant contributor to subsequent brain damage [1]. Therefore, managing patients with neurocritical illness requires monitoring and treating elevated ICP, especially in cases in children. In terms of precise and real-time measurements, invasive ICP measurements are presently the gold standard for the initial diagnosis and follow-up assessment of ICP [2]. Nevertheless, invasive procedures include a risk of consequences, such as infection and bleeding. Additionally, the majority of children with acute encephalopathy have no invasive ICP monitoring. Consequently, a bedside approach that can easily, reliably, and non-invasively detect ICP in children is required [2].

Invasive techniques can be replaced with some non-invasive techniques for measuring ICP, such as fundoscopy, tympanic membrane displacement, transcranial Doppler, optic nerve sheath diameter (ONSD), computed tomography (CT), and magnetic resonance imaging (MRI) [3]. Complications are not as likely to occur with these minimally-invasive treatments as they are with invasive ones. Recent studies on adults found that assessing ONSD using non-invasive imaging modalities such CT, MRI, and ultrasound can be utilized as a substitute approach to assess increasing ICP [4,5].

On the other hand, CT and MRI for ONSD measurements take a long time, are expensive, and frequently involve patient transportation. Due to their cheap cost and quick bedside procedure without the requirement for radiation exposure, ultrasound evaluations of ONSD may thus be a better alternative, especially in the case of unstable patients and when they need real-time ICP monitoring [6,7].

A point-of-care test is described as an examination performed close to the patient at the time of the consultation with instantaneous findings available to make quick and informed decisions regarding patient treatment [2]. As a rapid and non-invasive method to detect elevated ICP, point-of-care ultrasonography (POCUS) of ONSD has been proposed. The ability of bedside POCUS of ONSD to detect elevated ICP with excellent diagnostic test accuracy in adults has already been demonstrated [8,9].

Nonetheless, data on the relationship between POCUS of ONSD and ICP in children are hard to come by. Therefore, the purpose of this review is to point out the most recent findings from the pediatric published literature and briefly discuss what was assessed with ONSD ultrasound examination, and also to describe and discuss the diagnostic procedures that are available for optic nerve ultrasound appraisal.

## 2. Materials and Methods

We looked through the medical databases PubMed and Scopus. Additionally, to gain a broader perspective and comprehension of the problem, a preliminary generic Google search was also carried out. We typed in terms such as “ocular ultrasonography”, “ICP assessment”, “children”, “point-of-care ultrasound”, and “POCUS”. The search terms used in the text were either selected using the existing literature as a guide or were taken directly from related bibliographies. For further inclusions, manual searches of the bibliographies were also carried out. This review only included English full-length papers, case reports, or case series that dealt with the examination of the optic nerve by ultrasonography in children.

## 3. Results

### 3.1. Ocular Signs of Increased ICP

Papilledema, the enlargement of the optic disc related to elevated ICP from any cause, is frequently used to test for increased ICP. However, papilledema on fundoscopic examination is rarely useful in emergency scenarios when an acute increase in ICP occurs [10], since optic disc swelling in cases of elevated ICP takes time to develop [11].

ONSD has been correlated with ICP. In fact, the optic nerve is encased in the optic nerve sheath, which is connected to the intracranial subarachnoid space and surrounds the optic subarachnoid space [12]. When ICP rises, cerebral spinal fluid moves from the intracranial cavity into the optic subarachnoid space, causing the optic nerve sheath to swell and its diameter to expand [13]. As a result, the ONSD reflects variations in ICP, with a peak in the sheath distension at 3 mm after the papilla [14]. Moreover, an excessively increased ONSD may indicate elevated ICP before the onset of papilledema [14].

### 3.2. Ultrasound Technique

Generally, for B-scan ultrasonography, exams often use a high frequency and high-resolution (5–14 MHz) linear array probe. The patients are usually evaluated in a supine position.

To view the optic nerve insertion, most of the physicians gently place the probe on the closed eyelid with a typical ultrasonic gel and angled appropriately.

However, this may lead to errors in the assessment and measurement of ONSD, since the patient’s gaze direction is not visualized. For this reason, well-trained ophthalmologists usually perform this diagnostic examination with the lids open, using anesthetic drops and methylcellulose, thus making the examination more accurate [15,16,17].

After adjusting the B-scan probe to better visualize the optic nerve insertion (Figure 1), the optic disc is used as a reference point for measuring the ONSD diameter from inner edge to inner edge at a distance of 3 mm behind the globe.

In addition to B-scan examination, there is another ultrasound technique which is well known in ophthalmology: the “standardized echography”. It is a combination of A-scan, B-scan, and Doppler techniques that is usually applied with direct contact between the ultrasound probe and the surface of the eye [18]. The peculiarity of this diagnostic method lies precisely in the special design and standardization of the A-scan instrumentation. In fact, standardization permits every examiner to obtain the same echograms when evaluating the same structure, to achieve comparable and repeatable results [18]. Furthermore, the use of standardized ultrasonography is limited neither by the age nor by the general health of a patient. Nonetheless, infants between six months and three years may require sedation to allow for a complete and thorough examination [18]. Concerning the optic nerve, its thickness can be measured by standardized echography with accuracy ranging from +0.3 mm in the retrobulbar space to +0.5 mm in the posterior orbit [18]. Moreover, this ultrasound technique can detect neural swelling, thickening of the sheaths, increased subarachnoid fluid, atrophy, and other pathological conditions such as meningioma and glioma [18,19].

Unfortunately, it should not be assumed that B-scan ultrasonography is a particularly objective technique for measuring the optic nerve. This ultrasound approach, which has been previously reviewed in the literature [20,21,22], has a number of limitations when evaluating small structures, such as the optic nerve, since the B-mode does not have a uniformity of the gain setting [23,24,25,26]. This phenomenon is known as the “blooming effect” (different from the Doppler one) [18,27]. For this reason, the standardized A-scan technique, which is a blooming effect-free method that uses an 8 MHz non-focused probe with a unique S-shaped amplification and allows for more accurate measurements, especially in the case of optic nerve evaluation, might solve this issue [28].

This technique can also circumvent the issue of caliper position and provides exacter and more precise results by demonstrating clearly-apparent high reflective spikes from the interface between the arachnoid and subarachnoid fluid [18].

Furthermore, it is possible to perform the “30-degree test” with the standardized A-scan examination [29]. With the use of this test, the physicians can distinguish between an increase in ONSD brought on by increased subarachnoid fluid related to elevated ICP and that which is brought on by other disorders such as optic neuritis or optic nerve meningioma. In healthy and cooperative patients, this maneuver is performed with the patient looking straight ahead and then to the lateral side; in this way, intracranial hypertension caused by increased subarachnoidal fluid, which causes ONSD distension, will be demonstrated if this test shows a decrease in the maximal diameter of at least 5%. In uncooperative patients, such as comatose or anesthetized patients, the same test could be performed using forceps to mobilize the eye globe [30].

In addition, to rule out an optic nerve compartment syndrome, it is also possible to perform the “optic nerve exercise” test. In this test, the patient is asked to alternately glance to the very right and left lateral sides for 15 to 20 s. After the test, which lasts three minutes, the patient is instructed to close their eyes for two minutes to enable the subarachnoid fluid, which was forced out of the orbit during the exercise, to return. The orbital subarachnoid fluid will often return to its initial level in healthy individuals; however, it will not return in cases with optic nerve compartment syndrome [31].

### 3.3. Optic Nerve Ultrasound Evaluation in Healthy Children

The first step to try to detect pathological conditions through ocular ultrasound in children is to find the baseline values of ONSD in healthy subjects. Several papers have been published in the scientific literature on the use of B-scan to appraise the optic nerve in healthy children.

Lan et al. [32] retrospectively reviewed 250 full-term neonates and found that male neonates had a larger ONSD than female neonates (3.34 +/− 0.22 mm versus 3.26 +/− 0.20 mm). Fontanel et al. [33] created an optic nerve growth curve from normal ONSD values in subjects from 0 to 18 years of age and also identified age-appropriate ONSD cut-off values to be used in the diagnosis of intracranial hypertension. They calculated two different cut-off values of ONSD based on age groups: for the 4–10 years group, the ONSD cut-off was estimated as 4.10 mm and, for 11–18 years, the OSND cut-off was estimated as 4.4 mm. These values are comparable with the values found by Maude et al. [34] and Ballantyne et al. [35] (ONSD range: 2.1–4.3 mm, cut-off 4 mm in infants less than 1 year and 4.5 mm in older children), although they are not comparable with those found by Steinborn et al. [36], who found a mean ONSD of 5.75 mm. Further studies are needed to evaluate the correct cut-off values for these age ranges.

Furthermore, ONSD ultrasound measurement is also feasible in fetuses, and Haratz et al. [37] found an ONSD increase from 1.2 mm at 23 weeks to 2.6 mm at 36 weeks. In addition, ONSD enlargement was also observed in fetuses with intracranial lesions, which may be a useful tool to diagnose increased ICP early [37]. However, further studies are also needed in this case to better standardize the examination procedure.

### 3.4. Optic Nerve Ultrasound Evaluation in Children with Intracranial Hypertension

Many distinct neurological conditions, such as trauma, infection, hydrocephalus, intracranial mass, vasculitis, or idiopathic conditions, can cause increased ICP. For the purpose of preventing brain impairment and the associated death, early identification and treatment are crucial, especially in children. Due to its reproducibility and non-invasive procedure, ONSD evaluation with B-scan has become more and more common in recent years, including in the pediatric setting. In fact, several papers have been published in the scientific literature concerning the use of B-scan ultrasonography in children with increased ICP.

In Table 1, the main papers on this challenging topic are summarized.

In all of the papers [14,38,39,40,41,42,43,44,45,46,47,48,49,50,51,52,53,54,55,56] shown in Table 1, the authors evaluated children with various conditions causing increased ICP, all using B-scan ultrasound. Although good sensitivity and specificity were also demonstrated in children, all of the papers show enormous variability in the proposed reference values, making standardization of the method difficult. Therefore, further studies in children with intracranial hypertension are needed to try to standardize the diagnostic ultrasonographic procedure and to identify accurate and fixed reference values.

### 3.5. Optic Nerve Ultrasound Evaluation in Children with Head Trauma

Head trauma is a very dangerous and life-threatening clinical condition that needs to be evaluated as soon as possible. Considering that head trauma often results in ICP increase, ONSD evaluation with ocular ultrasound can be considered an excellent diagnostic means to check for changes in ONSD that might reflect changes in ICP variations.

Some published papers examined the role of this diagnostic tool in children with traumatic brain injury [57,58,59,60,61].

All of these studies were performed utilizing B-scan ultrasonography and confirmed that ONSD increased together with ICP in children with traumatic brain injury, also showing a good correlation. For this reason, ONSD ultrasound evaluation may be considered to be a useful tool for assessing ICP where invasive monitoring is unavailable or contraindicated. However, also in this case, each study provided different cut-off values, confirming the need for standardization of the diagnostic procedure to obtain more accurate, objective, and repeatable values that can be used as reference values.

### 3.6. Optic Nerve Ultrasound Evaluation in Children Undergoing Surgical Procedures

Ocular ultrasound could also be considered to be an excellent diagnostic tool to monitor any changes in ICP during surgical procedures, including the ones carried out on pediatric patients. In fact, several papers published in the literature have addressed this challenging topic.

Adverse neurological consequences may occur following cardiac surgery and cardiopulmonary bypass (CPB). In fact, cardiac surgery and CPB may alter the blood–brain barrier, autoregulation, and ICP in the immediate postoperative period. In a study cohort including 14 patients [62], ONSD was measured with ocular ultrasonography after anesthetic induction and endotracheal intubation, and after separation from CPB. Due to increased systemic venous pressure and lower cardiac output, patients with Fontan physiology may be more susceptible to having higher baseline ICP. As an alternative, high central venous pressures could artificially inflate ONSD without causing clinical ICP changes, necessitating distinct normative values based on the kind of congenital heart disease.

In children, the elevated ICP caused by craniosynostosis (CS) is extremely important. For this reason, the use of bedside ocular ultrasonography to evaluate ONSD variations in CS children has started to rise [63]. In 30 infants enrolled for CS correction, hemodynamic parameters, airway peak pressure, oxygen saturation, and end-tidal carbon dioxide (ET CO_2_) were monitored. According to this study, prone positioning with head extension did not increase ICP, since no significant ONSD variations were found [64]. The ICP fluctuation in syndromic CS was also explored by simultaneous assessments of the ONSD and invasive 24-h intracranial pressure monitoring of five young patients with CS. The ONSD rose in all patients during the second portion of the night, and three patients showed elevated ICP, thus showing similar ONSD fluctuations throughout the course of the night in these children [65].

Moreover, a group of 128 children with syndromic or complicated CS were assessed for elevated ICP via fundoscopy to detect papilledema. In order to determine the validity and usefulness of measuring the ONSD using ultrasonography, the measurements were compared to ONSD determined by CT scan and concurrent fundoscopy. They concluded that, even if the ONSD is larger in children with papilledema, fundoscopy is still the only option for daytime screening [66].

It has also been proven that the use of a mouth gag, used during oral and neck surgery, causes significant increases in ONSD measurements of children that continue to increase 20 min after gag placement, due to hemodynamic changes and a sympatho-adrenergic response similar to that seen in laryngoscopy [67,68].

It has also been demonstrated that different anesthetic procedures might result in ICP alterations that are also reflected in ONSD variations. In a prospective double-blind randomized trial [69], ICP was assessed in 40 children using ultrasound ONSD as a marker. Epidural analgesia was administered via bolus or infusion. In comparison to continuous infusion, lumbar epidural bolus did not enhance the likelihood of an increase in intracranial blood pressure. In another two studies [70,71], the same authors analyzed ONSD and its correlation with ICP in 72 children that were awake, sedated, or under general anesthesia. Although the influence of age and fontanelle status must be considered, transorbital ultrasonography assessment of ONSD is a reliable non-invasive approach to identifying elevated ICP in children in every clinical condition. With a highly-satisfying level of diagnostic accuracy, ONSD thresholds offer qualitative first direction regarding ICP categories.

For postoperative analgesia in children, caudal block is frequently used. Ocular ultrasound was used to evaluate the ONSD in 80 children to assess the effects of caudal block on ICP in relation to the volume of local anesthetic [72]. ICP can rise more during caudal block with a large volume of local anesthetic than during caudal block with a low volume of local anesthetic. However, 1.0 mL kg^−1^ of local anesthetic used for caudal block might also cause a considerable rise in ICP, which might reflect in ONSD increase [72]. Another interesting study measured ultrasound ONSD in 25 children having laparoscopic surgery to determine the extent of the increased ICP caused by carbon dioxide (CO_2_) pneumoperitoneum in all phases of anesthesia and surgery [73]. Compared to ONSD during anesthesia induction, ONSD increased considerably during CO_2_ pneumoperitoneum [73].

In a study carried out on pediatric and adult patients with hydrocephalus, ONSD changes using ultrasonography before and after the placement of ventriculoperitoneal shunt (VPS) were evaluated. After the shunt was implanted, a considerable decline in ONSD in both the adult and child populations was observed [74].

Another study on 32 children evaluated the accuracy of ONSD as a marker for VPS shunt failure, showing an ONSD greater than 4.0 mm in under-12-months children and greater than 4.5 mm in over-12-months children [75]. The same findings were also found by Hall et al. in a study on 39 children [76].

Choi et al. [77] evaluated ONSD in 34 children with hydrocephalus before and after VPS surgery. Before surgery, the mean ONSDs were 5.4 mm in the right eye and 5.3 mm in the left eye. Afterward, they were 4.4 mm in the right eye and 4.5 mm in the left eye. They concluded that, using this method, ICP can be quickly and painlessly assessed to determine the best postoperative strategy [77].

In acute liver failure (ALF), an increase in ICP brought on by cerebral edema plays a significant role in morbidity and mortality. A case study [78] showed the use of ocular ultrasonography to assess ICP in a two-year-old patient with ALF undergoing liver transplantation. Since invasive ICP monitoring was problematic given the severity of the coagulopathy, ultrasound was used to measure ONSD. ICP > 20 mmHg in this child was defined as a value of 4.5 mm, and this value was evaluated frequently throughout the procedure. Thus, also in patients with severe coagulopathy and high ICP after liver transplantation, ultrasonographic ONSD assessment could be a helpful method [78].

Finally, Padayachy et al. [79] investigated the connection between invasively-determined ICP in children and the measurement of the ONSD. The ICP reading was compared with the mean binocular ONSD measurement. Over the total patient group, composed of 174 children, 5.5 mm was the ONSD measurement with the highest diagnostic adequacy for identifying an ICP of 20 mmHg. A reliable and repeatable method for measuring the OSND using transorbital ultrasound has been shown to have a good correlation with ICP and excellent diagnostic specificity for identifying elevated ICP. The same authors [80] examined the diagnostic efficacy of age-related ONSD cut-off values in the same 174 children for detecting elevated ICP as well as the merit of employing anterior fontanelle patency to describe a distinct set of cut-off values. Before conducting an invasive ICP test on children under general anesthesia, the ONSD measurement was carried out. This was subsequently examined in children older than a year and younger than a year, and the anterior fontanelle was evaluated as a trustworthy physiological marker. Particularly at higher thresholds of 20 and 15 mmHg, transorbital ultrasound measurement of the ONSD is a valid non-invasive marker of ICP, and the authors also concluded that an important clinical marker for determining various ONSD cut-off values in children is the patency of the anterior fontanelle.

### 3.7. Optic Nerve Ultrasound Evaluation in Children with Systemic Disorders

Increased ICP in children may be not only an isolated condition, but also associated with the presence of a systemic pathology. For this reason, the use of ultrasound in patients with systemic pathology associated with increased ICP is very relevant, including in pediatric patients.

In all of the papers analyzed on this topic, the ultrasound examination was performed with a B-scan probe and the ONSD was analyzed 3 mm posterior to the globe. No paper utilized standardized A-scan ultrasound for the ONSD assessment.

Several studies have assessed the ONSD value in Diabetic Ketoacidosis in children with Diabetes Mellitus Type 1 [81,82,83,84,85,86].

The usefulness of this method in the management of patients with Diabetes Mellitus Type 1 complications is undisputed, especially in assessing the possible presence of cerebral edema.

However, not all studies agree on its usefulness in monitoring the treatment of diabetic ketoacidosis. Indeed, Kendir [82] et al. and Şık et al. [81] found a good correlation between ONSD measurements and ketoacidosis fluid treatment, while Hansen et al. [83] found no significant difference.

Similarly, Bergman et al. [84] stated that ONSD did not vary significantly according to the severity of the diabetes.

Among the studies that have examined ONSD in patients with ALF [87,88,89], Das et al. [88] agreed that values greater than 4.55 mm could suggest an ICP increase, while the cut-off of 5.1 mm is the limit beyond which special clinical attention is needed.

The ONSD assessment in patients with Plasmodium falciparum infection is a method that has proven to be useful and may provide important information on clinical conditions, especially in underdeveloped countries where access to treatment is difficult [90,91].

In Table 2, the main findings of the papers related to the use of ONSD ultrasound evaluation in children with systemic disorders are summarized.

As with the previously-discussed topics in this review, the authors [81,82,83,84,85,86,87,88,89,90,91,92,93] used only the B-scan technique to perform the ONSD assessment on children. In addition, there are very few published papers nowadays on the use of ONSD ultrasound to evaluate effects related to systemic diseases in children. Therefore, further studies with larger sample sizes are needed to better investigate the usefulness of this diagnostic method.

## 4. Discussion

In children who are neurocritically ill, increased ICP is a common consequence. Current diagnostic techniques are invasive and have several pitfalls. For this reason, POCUS of ONSD could be considered a safe, noninvasive bedside diagnostic tool that enables the real-time evaluation of elevated ICP and the early beginning of therapy, and may close this gap. Despite not being a novel technique, it seems to have untapped potential in pediatric neurocritical care. The most significant drawback at this time is the significant variation in POCUS of ONSD that has been documented across pediatric research in children with normal, healthy circumstances and children with pathological situations, as revealed by the conflicting results of cut-off values derived from the published scientific literature. The main reason for this issue may be the lack of standardization of the diagnostic procedure for B-scan, which is the most widely used by clinicians for optic nerve assessment.

In fact, as previously discussed in this review, in all of the included papers [14,32,33,34,35,36,37,38,39,40,41,42,43,44,45,46,47,48,49,50,51,52,53,54,55,56,57,58,59,60,61,62,63,64,65,66,67,68,69,70,71,72,73,74,75,76,77,78,79,80,81,82,83,84,85,86,87,88,89,90,91,92,93], the B-scan ultrasonography was utilized, which is known to be affected by the “blooming effect”, which is related to an absence of a standardized gain setting. For this reason, the caliper position to perform ONSD measurements could also be affected by bias which could alter the data reliability and objectivity. Moreover, in all of these papers [14,32,33,34,35,36,37,38,39,40,41,42,43,44,45,46,47,48,49,50,51,52,53,54,55,56,57,58,59,60,61,62,63,64,65,66,67,68,69,70,71,72,73,74,75,76,77,78,79,80,81,82,83,84,85,86,87,88,89,90,91,92,93], the B-scan examination was performed with closed eyelids, possibly leading to errors in the assessment and measurement of ONSD, since the patient’s gaze direction was not clearly visualized.

For all of the aforementioned limitations of the B-scan technique, the use of the standardized A-scan technique [15,16,17,18,19,20,21,22,23,24,25,26,27,28,29,30,31] coupled with B-scan should be suggested to provide data that are as accurate, precise, repeatable, and objective as possible. In this way, a complete ultrasound examination can be performed in which useful clinical information is not missed and bias is greatly reduced, resulting in more reliable results.

## 5. Conclusions

In conclusion, clinical trials in children in which standardized A-scan is also used are needed, to compare results not only with B-scan, but also with other noninvasive procedures and with invasive ICP monitoring, which, nowadays, continues to be the gold standard to determine ICP. In addition, we would also like to recommend that evaluations of elevated ICP values should be performed in conjunction with available information from clinical exams and other diagnostic modalities, and correlated with them, rather than only using ultrasound ONSD.

## Figures and Tables

**Figure 1 diagnostics-13-00535-f001:**
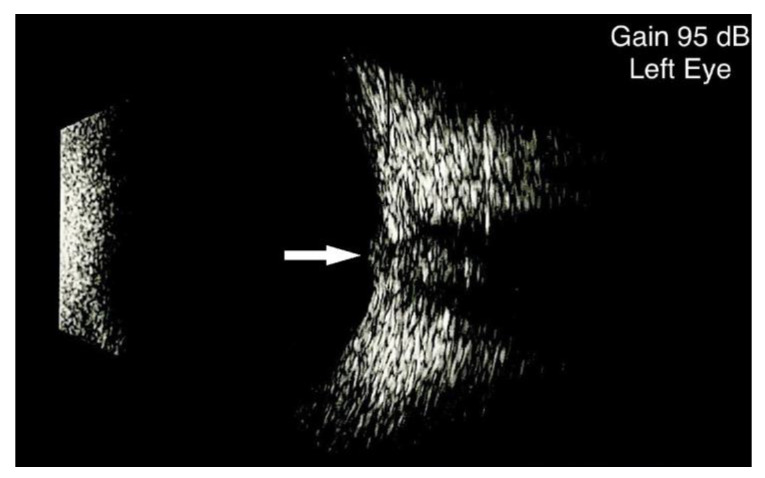
Optic nerve B-scan ultrasound in a child in which the optic nerve head is visible (white arrow). the presence of the optic nerve head in the ultrasound scan at the point where it emerges at the level of the retina, with no interposed scleral tissue, is a prerequisite for making ONSD measurements with B-scan ultrasonography.

**Table 1 diagnostics-13-00535-t001:** Main findings of the papers related to ONSD ultrasound evaluation in children with intracranial hypertension.

Author	Purpose of Study	N° of Patients	Methods of Evaluation	Results
Sarigecili et al. [38]	To compare the ONSD values between migraine patients with and without arachnoid cysts and to assess the correlation between the clinical severity of the cysts and their size on MRI.	64 patients divided into three groups: - 24 patients with migraine and arachnoid cysts - 20 with only headache without arachnoid cysts - 20 healthy controls	Optic nerve ultrasonography and brain MRI.	The size of the arachnoid cyst and the ONSD measurement were shown to be only marginally positively correlated Group 1 had the highest and lowest values for the ONSD.
Tessaro et al. [39]	To evaluate optic disc elevation by measuring ONSD POCUS in children with elevated ICP.	76 eyes from 40 children.	A blinded POCUS expert measured optic disc elevation, optic disc width at mid-height and ONSD.	ICP had increased in 26 subjects. With a sensitivity of 96% and a specificity of 93%, the ideal optic disc elevation of both eyes cut-off was 0.66 mm.
Arslan et al. [40]	To evaluate the relevance and any potential relationships between Near Infrared Spectroscopy and ultrasound ONSD in children with elevated ICP.	36 children: - 6 under the age of one year - 14 in the one to ten years age - 16 over ten years of age	Optic nerve ultrasonography and Near Infrared Spectroscopy.	The mean values of ONSD and Near Infrared Spectroscopy were respectively: - 4.8 ± 0.7 mm and 71.1 ± 12.4% in the first group - 6.1 ± 0.6 mm and 72.7 ± 9.3% in the second group - 5.6 ± 0.7 mm and 74.2 ± 16% in the third group. There was no correlation between the ONSD and Near Infrared Spectroscopy values
Biggs et al. [41]	To evaluate the possible correlation between ultrasound ONSD and elevated ICP in critically ill children.	16 children ≤ 18 years old.	13–6 MHz Linear ultrasound B-scan probe and invasively assessed ICP.	ICP and ONSD did not substantially correlate (*p* = 0.51).
Aslan et al. [42]	To assess the diagnostic use of central retinal artery Doppler indices and ONSD measurements in the assessment of pediatric patients with elevated ICP.	38 children with increased ICP and 19 healthy children.	Optic nerve ultrasonography and transcranial Doppler.	The mean ONSD was 5.9 mm in the study group and the mean resistive index (RI) was 0.71 ± 0.08 mm, significantly greater than the control group. The ONSD measurement was the strongest parameter to correlated with ICP.
Sharawat et al. [43]	To assess the diagnostic efficacy of transcranial Doppler-guided middle cerebral artery flow indices and ONSD in children, compared to the gold standard of invasive intraparenchymal intracranial pressure values.	30 children (2–12 years)	Optic nerve ultrasonography, Transcranial Doppler and invasive ICP monitoring.	The ONSD was found to have an excellent diagnostic accuracy in identifying children with an intracranial pressure of greater than or equal to 20 mm Hg, in contrast to transcranial Doppler-guided middle cerebral artery flow indices.
Aslan et al. [44]	To evaluate the correlation of the lumbar puncture opening pressure with the ultrasonographic ONSD and retinal resistive index measurements in patients with PTCS.	22 children: - 7 with PTCS - 15 healthy children	Optic nerve ultrasonography, transcranial Doppler and lumbar puncture.	A significant associations between the ONSD baseline measurements and the lumbar puncture opening pressure for both the right eye and the left eye was found. Opening pressure in lumbar puncture and retinal resistive index readings did not correlate. Ultrasonographic ONSD measurements can be utilized as a noninvasive method for PTCS patient follow-up and for initial ICP assessment.
Tekin Orgun et al. [45]	To establish the function of ONSD in the diagnosis and monitoring of young patients with idiopathic intracranial hypertension.	8 children.	Optic nerve ultrasonography.	Correlation between the mean ONSD (5.94 ± 0.46 mm) and the mean cerebrospinal fluid opening pressure (37.75 ± 12.64 cm H_2_O) was found.
Robba et al. [46]	To evaluate the interaction between ICP and various ultrasound-based techniques in pediatric patients requiring neurocritical care.	10 children	Optic nerve ultrasonography and Transcranial Doppler.	ONSD has the best correlation with ICP. According to the data, ONSD has an area under the curve of 0.94 when the threshold is 15 mmHg, with the optimal threshold being 3.85 mm (sensitivity = 0.811; specificity = 0.939).
Padayachy et al. [47]	To assess the deformability index’s diagnostic efficacy when used in combination with ONSD examination.	28 children (19 with high ICP)	Optic nerve ultrasonography	Patients with high ICPs had considerably lower deformability indexes than those with normal ICPs. Combining deformability index and ONSD evaluations increased association with ICP and diagnostic accuracy (sensitivity 94.7%, specificity 88.9%).
Rehman Siddiqui et al. [48]	To estimate ONSD using ultrasonography in the case of elevated ICP.	48 children with mean age of 7.5 ± 5.0 years	Linear ultrasound probe and trans cranial computed tomography scan/MRI.	Ultrasonographic ONSD measurement in babies, children and adolescents has been validated with a sensitivity and specificity of 100% and 60–66.7% respectively.
Marchese et al. [49]	To compare the feasibility and precision of POCUS with an ophthalmologist’s fundus examination in the detection of optic nerve anomalies related to edema in pediatric emergency medicine patients.	76 children	POCUS to measure ONSD and elevation of the optic disc.	The sensitivity and specificity were 90% and 55%, respectively, when using a sonographic definition for optic nerve swelling that included an optic nerve sheath diameter more than 4.5 mm or the presence of elevated optic disc.
Ozturk et al. [50]	To define the effectiveness of MRI and orbital ultrasonography in the diagnosis of idiopathic intracranial hypertension.	16 children: - 7 with idiopathic intracranial hypertension - 9 with pseudopapilledema	Optic nerve ultrasonography and MRI scans	ONSD was 4.62 ± 0.64 mm in pseudopapilledema patients and 6.62 ± 0.70 mm in papilledema patients. The cerebrospinal fluid opening pressure and ONSD showed a strong correlation.
Steinborn et al. [51]	To assess the precision of high resolution transbulbar sonography in estimating ICP in pediatric patients.	81 children.	Optic nerve ultrasonography	The ONSD was substantially higher in children with elevated ICP. Ultrasound ONSD could be considered an effective method to quickly and painlessly estimate ICP.
Irazuzta et al. [52]	To appraise the correlation between the ONSD and ICP in children with suspected idiopathic intracranial hypertension.	13 children (10 with idiopathic intracranial hypertension)	Optic nerve ultrasonography	Measurements that were deemed abnormal were those with an ONSD < 4.5 mm and a cerebrospinal fluid opening pressure > 20 cm H_2_O. In all patients, high ICP could be predicted or ruled out based on the ONSD.
Le et al. [53]	To examine the results for the sonographic measurement of the ONSD carried out by an emergency physician for the diagnosis of elevated ICP.	64 children (24 with idiopathic intracranial hypertension)	Optic nerve ultrasonography	ONSD had a sensitivity of 83% and a specificity of 38%; positive likelihood ratio of 1.32 (95% CI 0.97 to 1.79) and negative likelihood ratio of 0.46 (95% CI 0.18 to 1.23) for increased ICP were found. These results are insufficient to support medical decision-making.
McAuley et al. [54]	To evaluate transorbital ultrasound ONSD as a clinical assessment marker of developing hydrocephalus in the pediatric population.	160 children	Optic nerve ultrasonography	Ultrasound ONSD measurements were compared to clinical case information from related case files. This approach is regarded as a helpful adjuvant in the assessment of hydrocephalus.
Beare et al. [55]	To establish normal ONSD data for African children and assess optic nerve sheath ultrasound as a non-invasive means of identifying elevated ICP.	14 children	Optic nerve ultrasonography	The mean ONSD was 5.4 mm (range 4.3–6.2 mm) in children with ICP, while in children without neurological disease was 3.5 mm (range 2.5–4.1 mm). ONSD ultrasonography could be considered a reliable way to identify elevated ICP.
Newman et al. [14]	To define the value of ultraosound ONSD in children with shunted hydrocephalus who may have elevated ICP.	23 children with shunted hydrocephalus: - 6 with increased ICP - 17 with symptoms suggestive of intracranial hypertension.	Optic nerve ultrasonography	In patients older than 1 year old, the ONSD has an upper limit of 4.5 mm whereas in children younger than 1 year old, it is 4.0 mm. The average diameter of the optic nerve sheath was 5.6 mm in individuals with elevated ICP. These findings support that the ONSD could be used for monitoring and assessing children with hydrocephalus who may have elevated ICP.
Helmke et al. [56]	To correlate the ultrasound ONSD and the acute conditions of intracranial hypertension.	39 children admitted to the intensive care unit	Optic nerve ultrasonography	When compared to normal data, the ONSD in intensive care unit patients with elevated ICP ranged up to 6.8 mm and was noticeably larger.

ONSD: optic nerve sheath diameter; MRI: magnetic resonance imaging; POCUS: point-of-care ocular ultrasound; ICP: intracranial pressure; PTCS: pseudotumor cerebri syndrome.

**Table 2 diagnostics-13-00535-t002:** Main findings of the papers related to optic nerve sheath diameter ultrasound evaluation in children with systemic and ocular disorders.

Study	N° of Patients	Methods of Evaluation	Pathology	Results
Şık et al. [81]	43	Ultrasound B-Scan	Diabetic Ketoacidosis before and after treatment	ONSD decrease from the beginning of the treatment
Kendir et al. [82]	36	Ultrasound B-Scan	Diabetic Ketoacidosis	ONSD measurement could predict cerebral edema
Jeziorny et al. [86]	144 (40 as control)	Ultrasound B-Scan	Diabetic Ketoacidosis	ONSD may serve to predict risk of development of Cerebral Edema in patients with Type 1 Diabetes
Hansen et al. [83]	5	Ultrasound B-Scan	Diabetic Ketoacidosis	ONSD showed a peak during therapy, at resolution of acidosis, and at admission
Hansen et al. [85]	7	Ultrasound B-Scan	Diabetic Ketoacidosis	ONSD showed no significant differences during the treatment of Diabetic Ketoacidosis
Bergmann et al. [84]	108	Ultrasound B-Scan	Diabetic children	ONSD did not vary significantly based on Diabetes Mellitus Type 1-related illness severity
Vijay et al. [87]	46 (15 as control)	Ultrasound B-Scan	Pediatric acute liver failure	When ONSD > 4.55 mm, it is possible a clinically raised ICP
Das et al. [88]	88 (47 as control)	Ultrasound B-Scan	Pediatric acute liver failure	When ONSD > 4.55, it is possible a clinically raised ICP
Helmke et al. [89]	22	Ultrasound B-Scan	Hepatic Failure	ONSD trends could reflect the variations of ICP in hepatic encephalopathy
Beare et al. [90]	112	Ultrasound B-Scan	Cerebral Malaria and Malaria	ONSD was higher at the admission in patients who developed neurological sequelae
Murphy et al. [91]	33	Ultrasound B-Scan	Malaria	ONSD was increased in one third of all patients with malaria and in 100% of the patients diagnosed with cerebral malaria.
James et al. [92]	1 (case report)	Ultrasound B-Scan	Orbital Cellulitis	Ultrasound ONSD value of 5.2 mm, supposing an increase in case of Orbital Cellulitis
Schumacher et al. [93]	65	Ultrasound B-Scan	Mucopolysaccharidoses I, II and VI	Mean ONSD was thicker (5.35–6.71 mm)

## Data Availability

Not applicable.

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
