# Peer review of "Optic Nerve Ultrasound Evaluation in Children: A Review"

_diagnostics, 2023, doi:10.3390/diagnostics13030535_

Round 1

Reviewer 1 Report

This well prepared manuscript can be recommended for publication in its current form.

Author Response

Thank you very much for your comment.

Reviewer 2 Report

The paper was written-design and written. 

I have only two comments:

1. in the discussion, please be more precise, and describe more.

2. The tables are very useful, but the authors can add a figure to summarize their findings.

Author Response

The paper was written-design and written.

I have only two comments:

  1. in the discussion, please be more precise, and describe more.

RE: Thank you very much for your suggestion. We modified the Discussion section (pages 12-13).

  1. The tables are very useful, but the authors can add a figure to summarize their findings.

RE: Thank you very much for your suggestion. We added two small paragraphs after the two Tables, trying to point out the main findings and the main concerns of the discussed papers (page 8 and page 12).

Reviewer 3 Report

Guidance for Authors

The proposed review is very interesting and well-structured, even if some issues should be properly addressed.

Can you provide an example of a B-scan image for optic nerve insertion imaging?

Table 1 should be properly commented on in the text to help the reader navigate the studies included.

The discussion section should be extended highlighting the main limitations and drawbacks of the studies reviewed. Can you explain in detail the reason why standardized A-scan technique should be coupled with B-scan?

Finally, I suggest a general revision of the paper and explain all the acronyms (e.g., IIH).

Author Response

Guidance for Authors

The proposed review is very interesting and well-structured, even if some issues should be properly addressed.

Can you provide an example of a B-scan image for optic nerve insertion imaging?

RE: Thank you very much for your suggestion. We added the Figure 1 with its description (page 3).

Table 1 should be properly commented on in the text to help the reader navigate the studies included.

RE: Thank you very much for your suggestion. We added a small paragraph after the Table 1 (page 8).

The discussion section should be extended highlighting the main limitations and drawbacks of the studies reviewed. Can you explain in detail the reason why standardized A-scan technique should be coupled with B-scan?

RE: Thank you very much for your suggestion. We modified the Discussion section according to your comments (pages 12-13).

Finally, I suggest a general revision of the paper and explain all the acronyms (e.g., IIH).

RE: Thank you very much for your suggestion. We revised all the paper and we explained all the acronyms.